# Organisational Culture Research in Healthcare: A Big Data Bibliometric Study

**DOI:** 10.3390/healthcare11020169

**Published:** 2023-01-05

**Authors:** Xiaoping Qin, Richard Wang, Yu-Ni Huang, Jinhong Zhao, Herng-Chia Chiu, Tao-Hsin Tung, Jeff Harrison, Bing-Long Wang

**Affiliations:** 1School of Public Health, The University of Sydney, Camperdown, NSW 2006, Australia; 2Affiliation Program of Data Analytics and Business Computing, Stern School of Business, New York University, New York, NY 10012, USA; 3College of Medical and Health Science, Asia University, Taichung 41354, Taiwan; 4School of Health Policy and Management, Chinese Academy of Medical Sciences & Peking Union Medical College, Beijing 100730, China; 5Institute of Hospital Management, Tsinghua University, Shenzhen 518000, China; 6Evidence-Based Medicine Center, Taizhou Hospital of Zhejiang Province Affiliated to Wenzhou Medical University, Linhai 317000, China; 7Brooks College of Health, The University of North Florida, Jacksonville, FL 32224, USA

**Keywords:** culture, hospitals, big data, CiteSpace, VOSviewer

## Abstract

Across international healthcare, organisational culture and work environment have become central to all patient safety. However, there is a lack of comprehensive overview to assess and track the evolution of the literature on organisational culture in healthcare. This study aims to describe the current situation and global trends in organisational culture research in healthcare. The methodology is based on bibliometric mapping using scientific visualisation software (CiteSpace and VOSviewer). The big data were collected from the Web of Science core citation database. After applying the search criteria, we retrieved 1559 publications, which have steadily increased over the last two decades. In addition, 92 countries and regions have published studies on organisational culture in healthcare. The United States has made significant contributions to this field. In particular, organisational culture occupies an important position in the quality management of different types of care and caregiving. At the same time, organisational culture in healthcare may be inadequately researched in terms of theoretical underpinnings, which in turn leads to a lack of widespread dissemination of practice, and research on organisational culture in healthcare through evidence-based medicine may remain a significant focus and hot topic throughout the research field in the coming years.

## 1. Introduction

Organisational culture is an organisation’s core and soul [1]. Every organisation has its unique culture and, since the last century, many scholars and studies have developed an understanding of the impact of cultural aspects on organisational management. As a result, organisational culture has become a significant area of management theory. Scholars and managers have widely recognised the importance of organisational culture in the operation of organisations [2]. Nevertheless, it is undeniable that organisational culture is not a “superficial” phenomenon. On the contrary, it is “infused with symbols and symbolic meanings” [3] and is “undetectable in most cases” [4]. In other words, it is more than just a “way of doing things” and a “style of dress.” Still, the organisation’s culture is reflected in the values, norms, and deep-rooted beliefs of the employees and is the basis for the operations and methods of doing business in the organisation [5]. Therefore, unique culture can be a source of competitive advantage for some organisations [6]. Moreover, the master in the field of organisational culture, Edgar Schein, also pointed out in his research that “culture determines and limits strategy” [7], which is a good indication of the importance of organisational culture in management.

Across international healthcare, organisational culture, and work environment have become central to all patient safety. Hospital organisational culture (HOC) is a term that has become synonymous with patient experience, satisfaction, mortality and morbidity [8]. One study suggests that healthcare organisations should pay particular attention to organisational culture because “the shared beliefs, values, and feelings within the organisation guide the perceptions and approaches to the work to be accomplished” [9]. Gershon and others’ research further explains Sovie’s statement, “If aspects of the organisational culture are ill-defined, frequently shifting, poorly communicated, not reinforced, and poorly supported administratively, both the employees’ collective perceptions and their behaviours (i.e., delivery of care, safe work practices, and teamwork) will be inconsistent” [10]. It is evident that organisational culture in healthcare is important for the “success” of healthcare organisations. Therefore, it is necessary to assess and track the evolution of the literature on organisational culture in healthcare to gain new insights and knowledge to improve issues in healthcare.

This study has several primary purposes; first, we provide a new way to view healthcare organisational culture areas and their associations by examining co-citation and co-occurrence data. Second, we connect our evolutionary analysis to a comprehensive future research plan, which may generate a new research agenda for healthcare leadership. Thus, this review focuses on illuminating the research frontiers and future roadmaps for organisational culture research in healthcare.

## 2. Materials and Methods

The bibliometric overview of this study describes the landscape and trajectory of change in the research field through a perspective on healthcare organisational culture from 1990 to 2021. The methodology used in this review is based on bibliometric mapping [11,12], a visualisation technique that quantitatively displays the landscape and dynamic aspects of the knowledge domain [13]. The data were collected from the Web of Science (WoS) core citation database. Two Java-based scientific visualisation software (CiteSpace and VOSviewer), developed by Chaomei Chen [11] and Van Eck and Waltman [14], were used to analyse the data.

### 2.1. Sample

The data for this study were retrieved from the Web of Science on September 28, 2022. Web of Science was chosen as the search engine because it is the most widely accepted and commonly used database for analysing scientific publications [15]. The terms “organisational culture”, “hospital culture”, “health care”, and “hospital” were used as search topics. The period was set from 1990 to 2021 (the starting year in the results is 1991 because no articles on organisational culture in healthcare were published in 1990).

A total of 1809 publications related to organisational culture in healthcare were identified. Publications before 1990 and after 2022 were excluded. In addition, articles, review articles, and early access articles were included in the study. Finally, to minimise language bias, we have excluded documents that were not published in English. Each publication in the WoS contains details, including the year of publication, author, author address, title, abstract, source journal, subject category, references, etc. The contents of the database were detailed before the bibliographic analysis was performed. For example, some authors present their names in different spellings when submitting articles, so the data must be viewed in detail and consolidated. There were 1559 publications included (Figure 1) and exported to the VOSviewer and CiteSpace software for analysis of the following topics: global publication trends, countries, journals, authors, research orientations, institutions, and the quality of publications.

### 2.2. Introduction to CiteSpace and VOSviewer

CiteSpace is a Java application, designed and produced by Professor Chaomei Chen, to visualise and analyse trends and patterns in the scientific literature. It was designed as a tool for visualising progressive knowledge domains. By using CiteSpace, we can see how major areas of research are being investigated through specific articles, and understand the most active frontier areas within research. The most critical articles and historical turning points in these areas are also available from the software [16].

VOSviewer is a software tool for building and visualising bibliometric networks. It was developed by Van Eck and Waltman [14]. In VOSviewer, metric networks can be visualised and analysed for factors including journals, researchers, or individual publications, and can be constructed based on citations, bibliographic couplings, co-citations, or co-authorship relationships [14].

## 3. Results

### 3.1. Global Publication Trends

#### 3.1.1. Global Trends

After applying the search criteria, we retrieved a total of 1559 articles. Figure 2a shows the number of articles increased from 2 in 1990 to 194 in 2021. To predict future trends in global publications, we used a logistic regression model to create a time profile of the number of publications throughout the year. In order to predict future trends, a linear regression model was used to create a time profile of the number of publications throughout the year, and the model fit curve for the growth trend is shown in Figure 2b. The trend in publication numbers was fitted well to the time curve as R^2^ = 0.9626. The R-squared value is an indicator of the degree of fit of the trend line. The value reflects the goodness of fit between the estimated value of the trend line and the corresponding actual data; the better the fit, the more reliable the trend line is [17]. It is also predicted that the number of publications in organisational culture in healthcare will grow to approximately 800 by 2035, based on the trend of the model, which is nearly a fourfold increase, compared to 2021.

#### 3.1.2. Contributions of Countries and Regions

Figure 2c,d shows the distribution world map of the top 10 countries of total publications. The United States contributed the most publications (596, 38.2%), followed by the United Kingdom (239, 15.3%), Australia (172, 11.0%), and Canada (138, 8.8%).

#### 3.1.3. Total Sum of the Times Cited

Among all included publications, the United States had the highest sum of the times cited (21,918), while the United Kingdom ranked second (7637), followed by Australia (3717), and Canada (3709), respectively (Figure 3a). Table 1 shows the detailed numbers.

#### 3.1.4. Average Citation Frequency

The United States had the highest average numbers of citations (36.78 times), followed by the United Kingdom (31.95 times), Sweden (30.41 times), and Canada (26.93 times), as shown in Figure 3b.

#### 3.1.5. H-Index

Total citations and h-index reflect a country’s publications’ quality and scholarly impact [18]. Figure 3c shows the h-index rankings, where the top ranking is the United States (h-index = 76), followed by the United Kingdom (h-index = 47), Canada (h-index = 35), and Australia (h-index = 33).

### 3.2. Analysis of Publication

#### 3.2.1. Journals

Figure 4a shows the top 20 journals in which publications on organisational culture in healthcare are located, with 72 articles published in the “BMC Health Services Research,” 38 in the “Journal of Nursing Management,” 31 in the “Journal of Advanced Nursing“ and “Health Care Management Review”, and 29 in the “Journal of Health Organisation and Management”.

#### 3.2.2. Research Orientation

The top 20 research orientations are shown in Figure 4b. The most common research orientation was nursing (417 publications), healthcare science services (373 publications), health policy services (293 publications), and public environmental occupational health (201 publications).

#### 3.2.3. Authors

The top 20 authors with the highest number of publications are shown in Figure 4c, with a total of 154 articles/reviews in the last decade, representing 9.87% of all literature in the field. Braithwaite from Australia has published 20 papers, followed by Shortell from the US, Mannion from the US, and Bradly from the United Kingdom with 10 papers. All researchers listed as authors were included in this term for analysis, regardless of their relative contribution to the study. It is worth noting that we included all authors in this study for analysis, regardless of their relative contribution to the study.

#### 3.2.4. Institutions

Figure 4d shows the top 20 institutions with the most publications. The University of California System had the highest number of publications, with 61 papers, followed by Harvard University (54 publications), then the US Department of Veteran Affairs, and the University of London (53 publications).

### 3.3. Co-Occurrence Analysis

A mapping of keywords regarding organisational culture research in the healthcare field; the nodes’ size represents the frequency, while the line between the nodes reflects the co-occurrence relationship. A total of 3329 keywords were included; some keywords with the same meaning that occurred at the beginning of the analysis using VOSviewer, such as “quality of health care”, and “quality of care” were merged. Finally, we attached the thesaurus file to VOSviewer and found 60 keywords that met the criteria. All keywords were grouped into 4 clusters: quality of care (blue cluster), leadership (green cluster), organisational culture (red cluster), and research (yellow cluster) (Figure 5).

The most prominent themes in the study of organisational culture in healthcare are as follows. In the “quality of care” cluster, the most used keywords were “organisational culture”, “patient safety”, “safety culture”, and “safety management”. The main keywords in the “leadership” cluster were “healthcare research”, “nursing”, “evidence-based practice”, and “mental health”. The main keywords in the “organisational culture” cluster were “leadership”, “quality improvement”, “implementation”, and “culture”. In the “research” cluster, prominent keywords were “quality of care”, “nurses”, “hospital”, and “job satisfaction”.

### 3.4. Burst Analysis

Eighteen burst terms of the time bar chart represent the evolution of the topic over time, showing the update and interaction of the literature. Figure 6 shows keyword highlighting sorted by starting year. The keyword that first became a research hotspot was “information system”, which appeared from 1996 to 2010. It was also the research hotspot with the most extended duration. The second keyword was ”focus group”, which appeared from 2001 to 2011, followed by “medical error”, which appeared from 2005 to 2008. The most recent burst keywords (from 2020) were “intensive care”, health policy”, “human resource management”, and “evidence-based medicine”. The keyword “quality of care” was the keyword with the shortest duration.

In order of intensity, “medical error” had the most vigorous intensity (strength = 6.01), followed by “primary care” (strength = 5.97), and “mental health” (strength = 5.24). The keyword “focus group” had the weakest intensity (strength = 2.12).

## 4. Discussion

### 4.1. Global Trends in the Healthcare Organisational Culture Field

Our study of health organisation culture (HOC) research illustrates the current and past global trends in publications, contributing countries, institutions, and research directions. The field of HOC research has evolved over the past decades. However, as this study shows, the number of publications has steadily increased yearly, with 92 countries and regions publishing in the field, suggesting that research focused on HOC research and providing in-depth knowledge will likely increase in the future.

### 4.2. Quality and Status of Global Publications

The main purpose of Figure 2 and Figure 3 is to show the countries with the highest number of publications and the highest quality of publications in the world by citation rate and h-index. We also find that the majority of the countries publishing are developed countries, but that developing countries are also catching up. Total citations and h-index reflect a country’s publications’ quality and scholarly impact [18]. According to our study, the United States ranked first among other countries in the total number of publications, citations, and h-index, making the most considerable contribution to global HOC research. The United Kingdom and Canada also contributed significantly, with respectable total citation frequency and h-index, especially the United Kingdom, which ranked second in average citation frequency. Nevertheless, some countries, such as Sweden, Canada, and Australia, also play an important role, considering their high average citation frequencies. It is worth noting that eight of the top ten countries in the ranking of essential contributors are developed countries, and two developing countries (Brazil and China). In North America and Europe, the main emphasis is on reducing costs, standardising and improving the efficiency of services, and improving the quality of work life and behaviour change. In most developing countries, process facilitation and service efficiency are the main objectives [19]. In developing countries, the study of organisational culture also has a guiding role for hospitals to improve the quality of care, and with economic development gradually catching up with the pace of developed countries, this study also plays a reference role in learning from the experience of developed countries with developing countries.

The journals “BMC Health Services Research”, “Journal of Nursing Management”, “Journal of Advanced Nursing”, “Health Care Management Review”, and “Journal of Health Organisation and Management” made extraordinary contributions and had the most research on HOC. From this, we can see that these journals are our primary sources of information regarding the latest developments in HOC.

The fact that almost all of the top 20 institutions are from the top five countries with the highest number of publications, with more than half of them located in the United States, again reflects the tremendous academic influence of the United States in this field. This study demonstrates the important role that these top-tier institutions play in improving a country’s scholarship. In addition, the top 20 authors represent research leaders who are likely to significantly impact the future direction of research. Therefore, more attention should be paid to their work to remain up-to-date with the latest developments in this field.

### 4.3. Research Focus on HOC

Keywords are an essential part of a research paper and contain the most vital information [20]. Systematic analysis of keywords in specific research areas provides a clear understanding of trends and hot spots in different research areas [21]. In addition, co-occurrence analysis is based on the number of joint publications, to evaluate the relationship between the identified keyword domains. Therefore, it is an effective method for predicting future trends and hotspots in research areas of interest [22]. According to our findings in this study, the number of publications related to HOC research multiplied in 2002. HOC research continues to grow dynamically, the field of hospital management plays an essential role, and the effective management of organisational culture is one of the critical ways to improve performance [23]. The results of this boom will, in turn, encourage more researchers to commit to the future of HOC research. Through bibliometric and visual analysis, researchers can get an overall impression of the leading countries, authors, institutions, partnerships, and academic impact of HOC. This information is available to give investigators as a guide so they can selectively access advanced knowledge and valuable findings according to their requirements. In addition, co-occurrence analysis can describe trends and research hotspots in the field, thus further inspiring researchers for topic selection, and helping funding agencies develop profitable investment plans.

In this study, ultimately, a total of four possible research directions were summarised: “Organisational Culture”, “Leadership”, “Quality of Care”, and “Research”. With the help of this network diagram, we can clarify future trends further. As shown in the co-occurrence diagram, the keywords ”organisational culture”, “patient safety”, “care”, “leadership”, “quality of care”, and “hospitals” are highlighted with larger icons that are almost evenly distributed among “Organisational Culture”, “Leadership”, “Quality of Care”, and “Research”. Thus, investment in and demand for high-quality research is necessary for the context of these four research directions.

From these four research directions, many points can be drawn for discussion in HOC research. First, HOC leadership and healthcare organisations are complex networks of many professional groups, departments, and specialists to improve the quality of services and organisational performance of the healthcare system. Therefore, building up certain aspects of management systems and culture is necessary. However, most healthcare organisations have difficulty doing this [24]. A talented leader can catalyse change in these areas in a healthcare organisation to remain successful in a changing competitive environment [25]. Meanwhile, other health administrators and mid-level managers have vital roles and responsibilities in healthcare change actions [26]. That is why it is crucial and necessary to study the issue of leadership in the future comprehensively.

Secondly, modern medical research on the quality of care has been around for more than 50 years [27]. Moreover, the quality of care in the HOC is more complex than previously thought. Some cultural influences, such as excellence in care delivery, ethical values, engagement, professionalism, value for money, cost of care, commitment to quality, and strategic thinking, were identified as critical cultural determinants of quality care delivery [28]. Our study supports the rationale for the frontier and focus of research on the quality of care in the co-occurrence diagram, where it can be seen that quality of care and patient safety are the key factors for quality improvement in healthcare organisations [29]. Furthermore, available research indicates a huge demand for and cost of healthcare worldwide. Still, disparities in limited resources and clinical practice have increased the interest in improving the quality of healthcare in many countries around the world, especially in developed countries such as the UK and the US, where improving the quality of healthcare is high on the national agenda [30].

Third, research. In regards to evidence-based medicine (EBM) and evidence-based management (EBMgt), Stephen M. Shortell has also stated in past research that there are two components necessary to improve the quality of medical care: the development of EBM and EBMgt, which can identify better clinical practices, and knowledge of how to put these into routine practice, while also defining organisational strategy, structure, and change management practices. When the content understanding of clinical practice (EBM) is effectively applied in an excellent organisational context (EBMgt), quality of care can be improved and developed sustainably [31]. Therefore, the appropriate use of EBM and EBMgt has a guiding role in the role of HOC and quality of care in research, and is one of the current and future research priorities.

### 4.4. Research Milestones and Future Research

Research on healthcare leadership over the past 30 years (1991–2021) was divided into several phases, based on the evolution over time from burst analysis. In the first phase, our study found that information systems have been a high burst in the overall research process for about 14 of the last 30 years. Hospital executives worldwide have recognised the importance of considering information technology (IT) as a strategic element, and studies have suggested that the ability to innovate with IT is critical to improving hospital performance and quality of care [32,33]. Organisational culture has also been shown to influence innovation capabilities as it affects attitudes toward knowledge acquisition and cross-functional learning [34]. Specifically, the information system is inextricably linked to an organisation’s ability to innovate and is defined as the ability to identify the value of new information, absorb it, and apply it for productive purposes. Therefore, managing IT knowledge and culturally solid communication channels contributes to implementing innovation, resulting in better returns, user satisfaction, reliability, and competitive advantage. 

In the second phase, we found that between 2010 and 2015, there was a specific focus on emergency care and primary care in the HOC field, as well as increased research on the quality of care and quality management, which confirms the importance of organisational culture in healthcare. Thus, our study demonstrates the importance of organisational culture on healthcare and quality of care, as discussed above.

In the third phase, we observed the healthcare research among the burst studies, the emergence of quantitative research, and the persistence of evidence-based medicine as a research hotspot in the HOC field after 2018. Due to the complexity of management research topics, researchers are typically required to employ a range of quantitative and qualitative data collection methods and analysis techniques, with methodological trade-offs, depending on the research questions driving the study, their prior work, the planned research design, and the desired contribution the researcher wishes to make [35]. Qualitative research is unique in its ability to solve descriptive, explanatory, and illustrative problems, while quantitative research is better suited to generalisation and calibration problems [36]. Qualitative and quantitative research each have advantages and disadvantages; qualitative research usually obtains theories through experience, processes, and causal mechanisms, while quantitative research extends theories to large populations by refining or calibrating the understanding of a phenomenon. When theories are not adequately covered, they are re-examined and reviewed using alternative methods [37]. From the results, we speculate that there are studies within the HOC field investigating the experiences, processes, and causal mechanisms of HOC starting in 2016. There is a wide range of extension practice research beginning in 2018. However, the heat only lasts for one year, indicating that theoretical practice research on healthcare material culture is still insufficient. Combining the discussion of EBM and EBMgt above in future research to maintain the development of qualitative and quantitative research can encourage more profound exploration and research on the development of HOC.

In the final phase, intensive care, health policy, human resource management, evidence-based medicine, and professionals remain at the forefront of hot topics within the HOC field. We figure out that these hotspot terms will continue to be popular in the coming years. For example, professionals became a hot topic in 2015, and the heat lasted six years. It is well known that the healthcare field requires a high level of professionalism and that healthcare professionals’ perceptions of HOC are essential, as these perceptions influence their recognition and trust in healthcare organisations and significantly impact performance [38].

### 4.5. Strength and Limitation

To the best of our knowledge, this is the first study to conduct a bibliometric analysis of healthcare organisational culture research. The bibliometric and visual analysis was used to identify hotspots and emergent events across countries, authors, and institutions. However, this study inevitably has some limitations. Firstly, we only retrieved data from the WoS database since 1990. Therefore, we may have missed some publications due to database bias. Second, most of the identified publications were in English, and some articles related to other languages may not have been included. Third, some novel and high-quality articles with low citation frequency were not included in the study, due to the software base setting. So, there may be some bias in the study.

## 5. Significance

This study presents a bibliometric analysis of the current literature on organisational culture in healthcare. The study makes innovative use of two of the most popular software tools in bibliometrics to analyse the current English language literature published in the Web of Science. It provides an overview of the past and informs future research developments to improve the development of organisational culture as a core issue in healthcare management, especially hospital management, which is important for healthcare professionals around the world.

## 6. Conclusions

This study describes the current situation and global trends in organisational culture research in healthcare. The United States has made significant contributions to this field, establishing itself as a global leader. It is foreseeable that an increasing number of publications will be published in the coming years, which indicates the flourishing of organisational culture research in healthcare. In particular, organisational culture occupies an important position in the quality management of different types of care and caregiving, making it one of the central topics within the entire industry. At the same time, organisational culture in healthcare may be inadequately researched in terms of theoretical underpinnings, which in turn leads to a lack of widespread dissemination of practice, and research on organisational culture in healthcare through evidence-based medicine may remain a significant focus and hot topic throughout the research field in the coming years.

## Figures and Tables

**Figure 1 healthcare-11-00169-f001:**
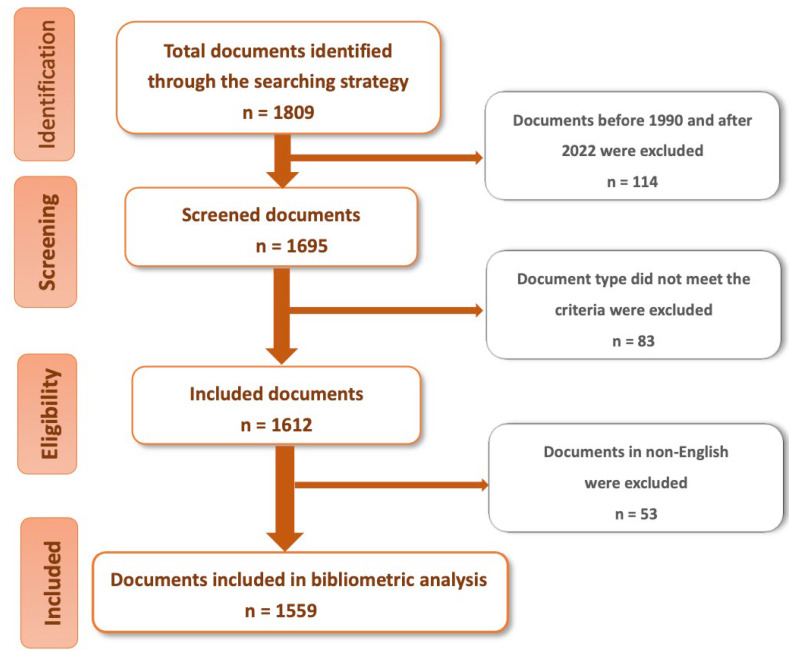
The research flow chart of the bibliometric analysis.

**Figure 2 healthcare-11-00169-f002:**
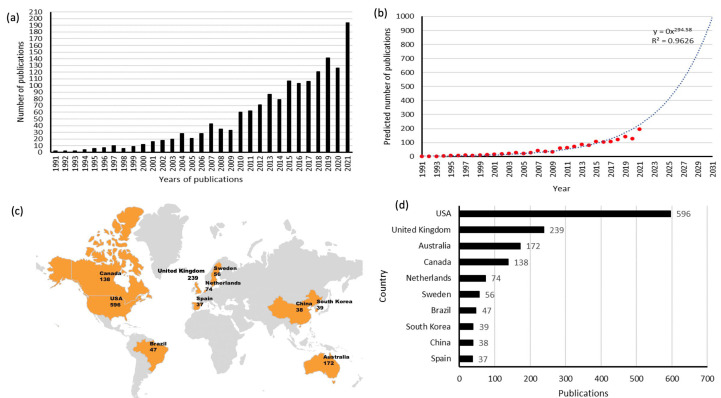
(**a**) The number of cumulative publications; (**b**) Model fitting curves of global publication trends; (**c**) The distribution world map of publications; (**d**) The top 10 countries of total publications.

**Figure 3 healthcare-11-00169-f003:**
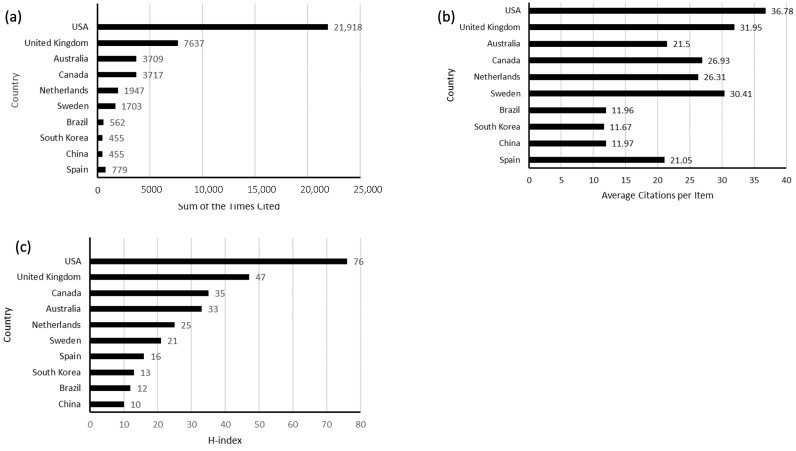
(**a**) The top 10 countries of total citation frequency; (**b**) The top 10 countries of average citations for each article; (**c**) The top 10 countries of the h-index.

**Figure 4 healthcare-11-00169-f004:**
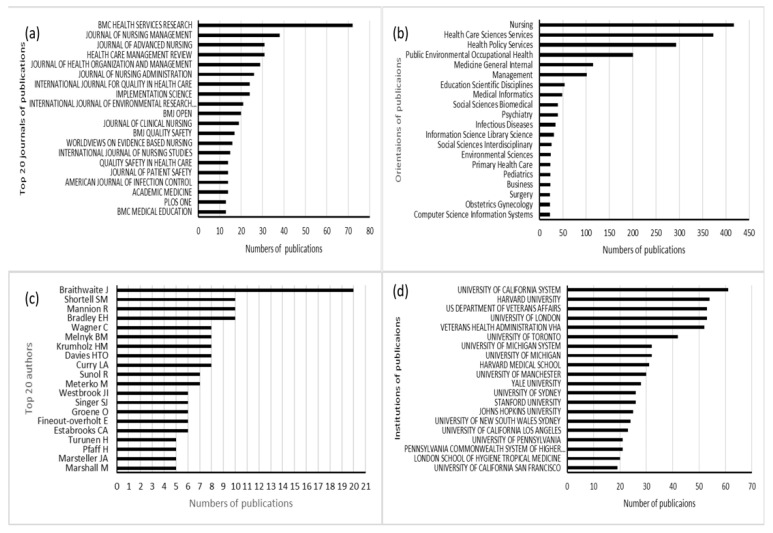
(**a**) The top 20 journals of publications; (**b**) The top 20 orientations of publications; (**c**) The top 20 authors with the highest number of publications; (**d**) The top 20 institutions with the highest number of publications.

**Figure 5 healthcare-11-00169-f005:**
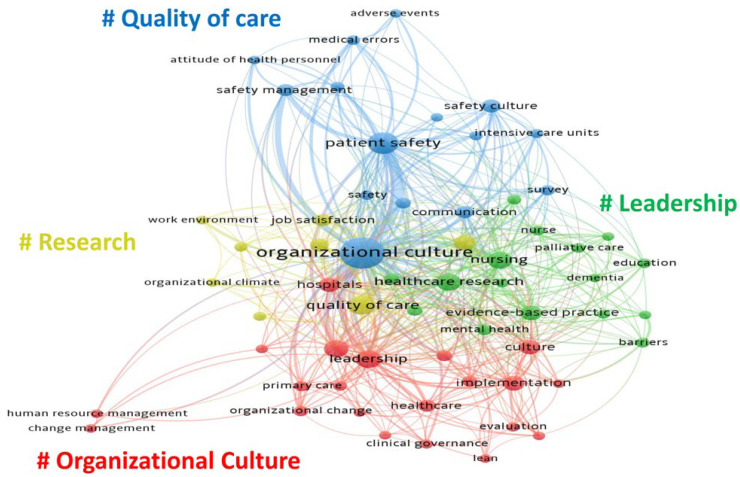
Co-occurrence analysis of organisational culture research in healthcare.

**Figure 6 healthcare-11-00169-f006:**
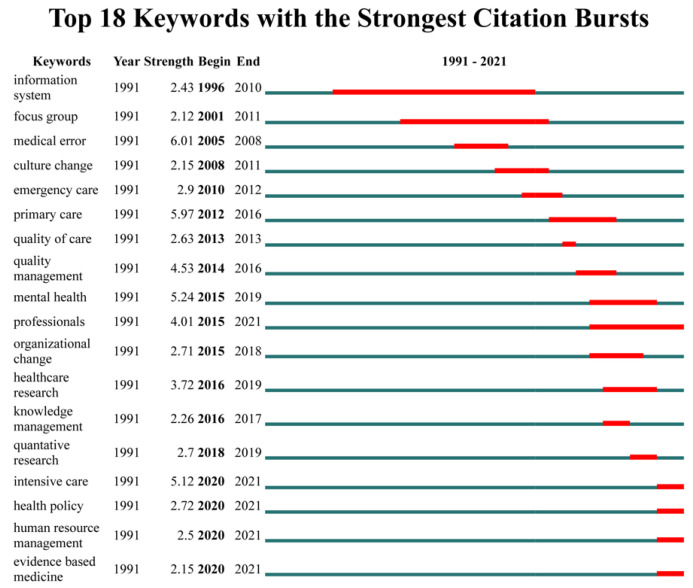
Temporal bar graph for burst terms.

**Table 1 healthcare-11-00169-t001:** The contributions in publications of countries.

Country	Publications	Sum of the Times Cited	Average Citations per Item	H-Index
USA	596	21,918	36.78	76
United Kingdom	239	7637	31.95	47
Australia	172	3709	21.5	33
Canada	138	3717	26.93	35
The Netherlands	74	1947	26.31	25
Sweden	56	1703	30.41	21
Brazil	47	562	11.96	12
South Korea	39	455	11.67	13
China	38	455	11.97	10
Spain	37	779	21.05	16

## Data Availability

Not applicable.

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
