# Peer review of "Organisational Culture Research in Healthcare: A Big Data Bibliometric Study"

_healthcare, 2023, doi:10.3390/healthcare11020169_

Round 1
Reviewer 1 Report
In this manuscript, the authors summarize 1,559 publications using bibliometric mapping with the scientific visualization software Citespace and VosViewer to explore the current state and global trends in health care organizational culture research. The manuscript is well-organized and clearly stated. I would suggest accepting it after the following major concerns are addressed.
(1)The second panel is Materials and Methods, but lacks an introduction to research methods. An introduction to Citespace and VosViewer should be added in panel 2.2.
(2)The fitted time curve in the fifth row of part 3.1.1 of the results should be R² instead of R2. Also what is the scientific basis for this R value?
(3)3.2.4 Part of the first and last sentences are repeated, please delete one sentence.
(4)What do the results in Figures 2 and 3 show? In addition to the specific values of the top four in words, can the remaining values be represented in the figure?
(5) In the discussion section, there are some logical inconsistencies and inconsistencies. According to the previous legend, the statement in the sixth line of section 4.2, the second highest total citation frequency and H-index would be the UK, not Canada as the authors write.
(6)There are some details in the article that need attention, especially the formatting. For example, there are problems with the formatting of part 3.1.2, so please check its line spacing and font boldness carefully.
(7)The significance of this paper is not expound sufficiently. The author need to highlight this paper's innovative contributions.
(8)The reference documents are of a long age, which cannot be reflected by the latest standards and methods.
Reviewer 2 Report
The aim of the research was to describe the current situation and global trends in research on organisational culture in healthcare. The research involved a bibliographic analysis. Articles of the time frame 1990 to 2021 were selected for it. I congratulate the authors for their determination and the extent of the research. The research process was conducted properly. A selection of areas for future research was made. What is missing is an inference as to why the US, UK and Australia were noted to have the highest number of publications on the topic under study. It is also worth supplementing the materials and methods subsection with a more detailed presentation of the Java method used (CiteSpace and VOSviewer).
Reviewer 3 Report
Your study is a novel approach to bibliographic big data analysis over a 10-year time period looking for organizational culture issues, assuming organizational culture is central question in health care quality and patient safety
It’s well written and worth’s reading
There are some questions:
Why was the top limit established in 2010? 12 years ago, from now
Line 203 why "future"? is it not current?
Round 2
Reviewer 1 Report
The response should be one to one point.
Author Response
Dear Reviewer,
Thank you for your helpful suggestions. We have uploaded a one-to-one point response document in our last response. Please check the attachment. If you have any other suggestions or comments, please let us know.
Sincerely.
Xiaoping
---
Response to Reviewer 1 Comments
Point 1: The second panel is Materials and Methods, but lacks an introduction to research methods. An introduction to Citespace and VosViewer should be added in panel 2.2.
Response 1: Thanks for your comments and suggestions. We have revised and added a panel 2.2. between lines 117-128.
2.2. Introduction to CiteSpace and VosViewer
CiteSpace is a Java application designed and produced by Professor Chaomei Chen to visualise and analyse trends and patterns in the scientific literature. It was designed as a tool for visualising progressive knowledge domains. By using CiteSpace we can see how major areas of research are being investigated through specific articles and understand the most active frontier areas within research. The most critical articles and historical turning points in these areas are also available from the software [16].
VOSviewer is a software tool for building and visualising bibliometric networks. It was developed by Van Eck and Waltman [14]. In VOSviewer, metric networks can be visualised and analysed for factors including journals, researchers or individual publications and can be constructed based on citations, bibliographic couplings, co-citations or co-authorship relationships [14]
Point 2: The fitted time curve in the fifth row of part 3.1.1 of the results should be R² instead of R2. Also what is the scientific basis for this R value?
Response 2: Thanks for your comments and suggestions.
(1)We have changed R2 to R2 and re-edited the sentence in line 136-139.
In order to predict future trends, a linear regression model was used to create a time profile of the number of publications throughout the year, and the model fit curve for the growth trend is shown in Figure 2b. The trend in publication numbers was fitted well to the time curve as R2=0.9626.
(2) The scientific basis description for R value has been added in line 139-142
The R-squared value is an indicator of the degree of fit of the trend line. The value reflects the goodness of fit between the estimated value of the trend line and the corresponding actual data; the better the fit, and the more reliable the trend line is [17].
Point 3: 3.2.4 Part of the first and last sentences are repeated, please delete one sentence.
Respond 3: Thank you for your suggestion, we have revised this paragraph in line 200-203
Figure 4d shows the top 20 institutions with the most publications. The University of California System had the highest number of publications, with 61 papers, followed by Harvard University (54 publications), then the US Department of Veteran Affairs and the University of London (53 publications).
Point 4: What do the results in Figures 2 and 3 show? In addition to the specific values of the top four in words, can the remaining values be represented in the figure?
Respond 4: Thanks to your comments. In the discussion section (lines 248-251) we have further explained Figures 2 and 3: The main purpose of Figures 2 and 3 is to show the countries with the highest number of publications and the highest quality of publications in the world by citation rate and H-index. We also find that the majority of the countries publishing are developed countries, but that developing countries are also catching up.
We have added the remaining values in Figures 2 and 3 in line 148 and 164.
Figure 2. (a) The number of cumulative publications;(b) Model fitting curves of global publication trends; (c) The distribution world map of publications; (d)The top 10 countries of total publications.
Figure 3. (a) The top 10 countries of total citation frequency; (b) The top 10 countries of average citations for each article; (c) The top 10 countries of the h-index.
Point 5: In the discussion section, there are some logical inconsistencies and inconsistencies. According to the previous legend, the statement in the sixth line of section 4.2, the second highest total citation frequency and H-index would be the UK, not Canada as the authors write.
Respond 5: We appreciate your suggestion and we have changed the error between lines 255-257.
The United Kingdom and Canada also contributed significantly with respectable total citation frequency and H-index, especially The United Kindom, which ranked second in average citation frequency.
Point 6: There are some details in the article that need attention, especially the formatting. For example, there are problems with the formatting of part 3.1.2, so please check its line spacing and font boldness carefully.
Respond 6: Formatting errors have been changed in lines 149-152.
Point 7: The significance of this paper is not expound sufficiently. The author need to highlight this paper's innovative contributions.
Respond 7: Thanks to your suggestion, we have added the paragraph of significance in lines 402-410.
- Significance
This study presents a bibliometric analysis of the current literature on organisational culture in healthcare. The study makes innovative use of two of the most popular software tools in bibliometrics to analyse the current English language literature published in the Web of Science. It provides an overview of the past and informs future research devel-opments to improve the development of organisational culture as a core issue in healthcare management especially hospital management, which is important for healthcare professionals around the world.
Point 8: The reference documents are of a long age, which cannot be reflected by the latest standards and methods.
Respond 8: Thanks to your suggestions, we have updated some of the relatively old references.
- Several of the references below have been replaced by literature that has the same statements in recent years.
- Gallagher RS. The Soul of an Organization: Understanding the Values that Drive Successful Corporate Cultures. Dearborn Trade 387 Pub.; 2003.
- Hartley J, Benington J. Leadership for healthcare. Policy Press Bristol; 2010.
- Moradi G, Sarbaz M, Kimiafar K, Shafiei N, Setayesh Y. The role of hospital information system on Dr Sheikh Hospital performance promotion in Mashhad. Health Information Management. 2008;5(2).
- Caccia-Bava Mdo C, Guimaraes T, Harrington SJ. Hospital organization culture, capacity to innovate and success in technology adoption. J Health Organ Manag. 2006;20(2-3):194-217. doi: 10.1108/14777260610662735.
- Edmondson AC, McManus SE. Methodological fit in management field research. Academy of management review. 2007;32(4):1246-64.
- Lee TW. Using qualitative methods in organizational research. Sage; 1999.
Here is the new literature after the replacement
- Inah EU, Tapang AT, Uket E. Organizational culture and financial reporting practices in Nigeria. Research Journal of Finance and Accounting. 2014;5(13):190-8.
- Boamah SA. Emergence of informal clinical leadership as a catalyst for improving patient care quality and job satisfaction. Journal of Advanced Nursing. 2019;75(5):1000-9.
- Lam L, Nguyen P, Le N, Tran K. The relation among organizational culture, knowledge management, and innovation capability: Its implication for open innovation. Journal of Open Innovation: Technology, Market, and Complexity. 2021;7(1):66.
- Yardley L, Bishop F. Mixing qualitative and quantitative methods: A pragmatic approach. The Sage handbook of qualitative research in psychology. 2017:398-413.
- Glesne C. Becoming qualitative researchers: An introduction. ERIC; 2016.
- Bluhm DJ, Harman W, Lee TW, Mitchell TR. Qualitative Research in Management: A Decade of Progress. Journal of Management Studies. 2011;48(8):1866-91. doi: https://doi.org/10.1111/j.1467-6486.2010.00972.x.
- We have retained the following relatively old literature because their statements in the field of health care or in the field of organisational culture are irreplaceable classics.
- Druckman D, Singer JE, Van Cott HP. Enhancing organizational performance. National Academy Press Washington, DC; 1997.
- Barney JB. Types of competition and the theory of strategy: Toward an integrative framework. Academy of management review. 1986;11(4):791-800.
- Schein EH. Organizational culture. vol 2. American Psychological Association; 1990.
- Sovie MD. Hospital culture--why create one? Nursing Economic$. 1993;11(2):69-75, 90.
- Gershon RR, Stone PW, Bakken S, Larson E. Measurement of organizational culture and climate in healthcare. JONA: The Journal of Nursing Administration. 2004;34(1):33-40.
- Campbell SM, Roland MO, Buetow SA. Defining quality of care. Social science & medicine. 2000;51(11):1611-25.
